# Monocyte Subsets and Serum Inflammatory and Bone-Associated Markers in Monoclonal Gammopathy of Undetermined Significance and Multiple Myeloma

**DOI:** 10.3390/cancers13061454

**Published:** 2021-03-22

**Authors:** Daniela Damasceno, Julia Almeida, Cristina Teodosio, Luzalba Sanoja-Flores, Andrea Mayado, Alba Pérez-Pons, Noemi Puig, Paula Arana, Bruno Paiva, Fernando Solano, Alfonso Romero, Sergio Matarraz, Wouter B. L. van den Bossche, Juan Flores-Montero, Brian Durie, Jacques J. M. van Dongen, Alberto Orfao

**Affiliations:** 1Translational and Clinical Research Program, Cancer Research Center (IBMCC, USAL-CSIC), Cytometry Service (NUCLEUS) and Department of Medicine, University of Salamanca and Institute of Biomedical Research of Salamanca (IBSAL), 37007 Salamanca, Spain; daniela_damasceno@usal.es (D.D.); jalmeida@usal.es (J.A.); amayado@usal.es (A.M.); albaperezpons@usal.es (A.P.-P.); smats@usal.es (S.M.); jflores@usal.es (J.F.-M.); 2Biomedical Research Networking Centre Consortium of Oncology (CIBERONC) (CB16/12/00400), Instituto Carlos III, 28029 Madrid, Spain; lcsanoja-ibis@us.es (L.S.-F.); npuig@saludcastillayleon.es (N.P.); bpaiva@unav.es (B.P.); 3Leiden University Medical Center, Department of Immunology, 2333 ZA Leiden, The Netherlands; c.i.teodosio@lumc.nl (C.T.); w.vandenbossche@erasmusmc.nl (W.B.L.v.d.B.); j.j.m.van_dongen@lumc.nl (J.J.M.v.D.); 4Institute of Biomedicine of Seville, Department of Hematology, University Hospital Virgen del Rocío of the Consejo Superior de Investigaciones Científicas (CSIC), University of Seville, 41013 Seville, Spain; 5Service of Hematology, University Hospital of Salamanca (CAUSA) and IBSAL, 37007 Salamanca, Spain; 6Regulation of the Immune System Group, Biocruces Bizkaia Health Research Institute, Hospital Universitario Cruces, Plaza de Cruces 12, 48903 Barakaldo, Spain; paula.aranaberganza@osakidetza.eus; 7Centro de Investigación Médica Aplicada (CIMA), Instituto de Investigación Sanitaria de Navarra (IDISNA), Clinica Universidad de Navarra, 31008 Pamplona, Spain; 8Hematology Service, Hospital Nuestra Señora del Prado, Talavera de la Reina, 45600 Toledo, Spain; fesora@sescam.jccm.es; 9Primary Health Care Center “Miguel Armijo”, Primary Health Care of Salamanca, Conserjería de Sanidad de Castilla y León (SACYL), 37007 Salamanca, Spain; jaromerof@saludcastillayleon.es; 10Department of Immunology, Erasmus University Medical Center, 3015 GA Rotterdam, The Netherlands; 11Centro del Cáncer Cedars-Sinai Samuel Oschin, Los Angeles, CA 90048, USA; bdurie@myeloma.org

**Keywords:** plasma cell neoplasms, MGUS, multiple myeloma, monocytes, FcεRI monocytes, tumor microenvironment, inflammatory cytokines, immunosenescence, bone markers

## Abstract

**Simple Summary:**

We investigated the distribution of different subsets of monocytes (Mo) in blood and bone marrow (BM) of newly-diagnosed untreated monoclonal gammopathy of undetermined significance (MGUS), smoldering (SMM) and active multiple myeloma (MM), and its relationship with immune/bone serum-marker profiles. Our results showed decreased production of BM Mo with decreased counts of classical Mo (cMo) in BM and blood of SMM and MM, but not MGUS. Conversely, intermediate and non-classical Mo were significantly increased in MGUS, SMM and MM BM. In parallel, increased levels of interleukin (IL)1β were observed in a fraction of MGUS and SMM, while increased serum IL8 was characteristic of SMM and MM, and higher serum IL6, RANKL and bone alkaline phosphatase concentrations, together with decreased counts of FcεRI^+^cMo, were restricted to MM presenting with bone lesions. These results provide new insights in the pathogenesis of plasma cell neoplasms and the potential role of FcεRI^+^cMo in normal bone homeostasis.

**Abstract:**

Background. Monocyte/macrophages have been shown to be altered in monoclonal gammopathy of undetermined significance (MGUS), smoldering (SMM) and active multiple myeloma (MM), with an impact on the disruption of the homeostasis of the normal bone marrow (BM) microenvironment. Methods: We investigated the distribution of different subsets of monocytes (Mo) in blood and BM of newly-diagnosed untreated MGUS (*n* = 23), SMM (*n* = 14) and MM (*n* = 99) patients vs. healthy donors (HD; *n* = 107), in parallel to a large panel of cytokines and bone-associated serum biomarkers. Results: Our results showed normal production of monocyte precursors and classical Mo (cMo) in MGUS, while decreased in SMM and MM (*p* ≤ 0.02), in association with lower blood counts of recently-produced CD62L^+^ cMo in SMM (*p* = 0.004) and of all subsets of (CD62L^+^, CD62L^−^ and FcεRI^+^) cMo in MM (*p* ≤ 0.02). In contrast, intermediate and end-stage non-classical Mo were increased in BM of MGUS (*p* ≤ 0.03), SMM (*p* ≤ 0.03) and MM (*p* ≤ 0.002), while normal (MGUS and SMM) or decreased (MM; *p* = 0.01) in blood. In parallel, increased serum levels of interleukin (IL)1β were observed in MGUS (*p* = 0.007) and SMM (*p* = 0.01), higher concentrations of serum IL8 were found in SMM (*p* = 0.01) and MM (*p* = 0.002), and higher serum IL6 (*p* = 0.002), RANKL (*p* = 0.01) and bone alkaline phosphatase (BALP) levels (*p* = 0.01) with decreased counts of FcεRI^+^ cMo, were restricted to MM presenting with osteolytic lesions. This translated into three distinct immune/bone profiles: (1) normal (typical of HD and most MGUS cases); (2) senescent-like (increased IL1β and/or IL8, found in a minority of MGUS, most SMM and few MM cases with no bone lesions); and (3) pro-inflammatory-high serum IL6, RANKL and BALP with significantly (*p* = 0.01) decreased blood counts of immunomodulatory FcεRI^+^ cMo-, typical of MM presenting with bone lesions. Conclusions: These results provide new insight into the pathogenesis of plasma cell neoplasms and the potential role of FcεRI^+^ cMo in normal bone homeostasis.

## 1. Introduction

Plasma cell (PC) neoplasms consist of a wide spectrum of end-stage antibody-producing B-cell tumors [1,2] that range from pre-malignant conditions such as monoclonal gammopathy of undetermined significance (MGUS) and smoldering multiple myeloma (SMM), to symptomatic multiple myeloma (MM) and PC leukemia [3].

Tumor PC control and growth kinetics in both MGUS and MM depend both on the intrinsic characteristics of neoplastic PC and their close interaction with the tumor microenvironment [4,5]. Thus, malignant BM PC in MM, and to a less extent also in MGUS and SMM, have the ability to modify their surrounding immune and bone microenvironment, interfere with immune surveillance and ultimately lead to bone resorption and lysis via direct local cell-to-cell and cytokine-mediated interactions between monocyte/macrophages, stromal cells, osteoclasts and osteoblasts [6,7]. Such interactions contribute to increased survival of malignant PC [8], promote drug resistance and induce local angiogenesis, all of which favor tumor growth, in parallel with activation of osteoclasts and osteolysis [6,7]. Thus, increased numbers in BM of M2-polarized macrophages have been reported in MM compared to healthy donors (HD), MGUS and SMM patients [9], in association with a poorer patient outcome [10]. In turn, upregulation of the RANKL-receptor activator of nuclear factor κ B ligand-pro-osteoclastogenic factor, in parallel to decreased osteoprotegerin (OPG) levels, have been described in BM of MM [11,12,13,14,15] and also MGUS patients [16], in association with an accelerated turnover of BM osteoblasts and increased bone alkaline phosphatase (BALP) and RANKL levels in serum of these patients [17,18]. These alterations might result from an increased differentiation of monocytes (Mo) into osteoclasts [19] associated with higher secretion of interleukin (IL)6 [20], CXCL12 [21,22] and RANKL by both BM stromal cells [23] and activated T cells [24]. Altogether, this leads to increased recruitment of osteoclasts to the BM endosteal niche, followed by an inhibition of its decoy receptor OPG, and locally increased bone resorption which is typically observed in MM.

The functionally altered interactions between Mo (and other immune cells) and both osteoblasts and osteoclasts in BM of MGUS, SMM and MM might become detectable in blood at already early phases of the disease, via redistribution of specific subsets of blood Mo associated with the increase of specific serum biomarkers. Thus, altered counts of both classical (c)Mo (vs HD) and Slan^+^ non-classical (nc)Mo (vs HD and MGUS) have been reported in the blood of MM [25] in association with a higher BM tumor load [26]. In addition, patients with MM have been reported to show strongly increased levels of pro-inflammatory cytokines in serum such as IL1β, IL10, TNFα [27] and IL6, together with an enhanced spontaneous ex vivo secretion of inflammatory cytokines (vs HD) by blood Mo [28]. Apart from BM Mo and macrophages, other immune cells have also been implicated in the pathogenesis of MM [4,29]. Among other cells, these include myeloid-derived suppressor cells (MDSC) [30,31,32], regulatory T cells (Tregs) [30,33,34,35,36], natural killer (NK) cells [37,38,39,40], T cells [41,42] and dendritic cells (DC) [43,44,45], which further supports a relevant role for the interaction between tumor PC and their immune/bone microenvironment in the pathogenesis of PC neoplasms.

Despite all the above, at present, there are still limited data about the distribution of distinct subsets of Mo in blood and BM of MGUS, SMM and MM, and the potential relationship between these alterations and other immune and bone-associated serum markers, which might contribute to better understand the clinical and biological differences between distinct diagnostic categories of these PC neoplasms and the underlying pathogenic mechanisms.

Here, we investigated in detail the distribution of different subsets of cMo, intermediate Mo (iMo) and ncMo in blood and BM of newly-diagnosed untreated MGUS, SMM and MM patients, in parallel to a large panel of immune and bone-associated serum markers. Our ultimate goal was to identify altered cellular and soluble immune/bone interaction profiles that are associated with distinct diagnostic categories of the disease (MGUS vs. SMM vs. MM) and that contribute to determining their different clinical behavior.

## 2. Materials and Methods

### 2.1. Patients, Controls, and Samples

A total of 128 aspirated BM and 220 peripheral blood (PB) EDTA-anticoagulated samples were studied. These included: (i) 15 normal BM (median age: 59 y; range: 31–83 y) from HD that underwent orthopedic surgery, and 97 normal PB (median age: 62 y; range: 32–92 y) samples recruited from the general population for a total of 107 HD (including 10 paired samples); (ii) 19 BM (median age: 70 y; range: 44–85 y) and 22 PB (median age: 67 y; range: 31–85 y) specimens from 23 MGUS patients (36 paired samples); (iii) 13 BM (median age: 71 y; range: 61–82 y) and 13 PB (median age: 73 y; range: 61–82 y) samples of 14 newly-diagnosed untreated SMM cases (26 paired samples); and, iv) 81 BM (median age: 71 y; range: 45–85 y) and 88 PB (median age: 72 y; range: 45–85 y) specimens from 99 newly-diagnosed untreated MM patients (140 paired samples) (Appendix A). Whole-body computerized tomography was used to evaluate the presence of osteolytic lesions at diagnosis. All samples were collected at diagnosis at each participating center—University Hospital of Salamanca (HUS; Salamanca, Spain); University Clinic of Navarra (Pamplona, Spain); and Hospital Nuestra Señora del Prado (Talavera de la Reina, Spain)- and processed locally within 24 h after collection (Appendix A). None of the patients included in the study had been previously diagnosed with a monoclonal gammopathy. All participants gave their informed consent to participate in the study in accordance with the guidelines of the local Ethics Committees and the Declaration of Helsinki.

### 2.2. Flow Cytometric Analysis of BM and PB Subsets of Monocytes

Immunophenotypic identification, enumeration and characterization of the distinct subsets of Mo present in BM and PB samples were performed using an 11-color antibody combination based on the TiMaScan^TM^ tube (Cytognos, SL, Salamanca, Spain) as backbone, and that consisted of the following reagents: CD45-pacific orange (PacO; clone HI30), CD62L-brilliant violet 650 (BV650; clone DREG-56), CD16-BV786 (clone 3G8), CD36-fluorescein isothiocyanate (FITC; clone CLB-IVC7), CD14- plus CD34-peridinin chlorophyll protein-cyanin 5.5 (PerCPCy5.5; clones MφP9 and 8G12, respectively), anti-Slan-phycoerythrin (PE; clone DD-1), CD117-PE-CF594 (clone YB5.B8), anti-HLADR-PECy7 (clone G46–6), CD64-allophycocyanin (APC; clone 10.1), and CD300e (IREM2)-APCC750, (clone UP-H2). In a subset of 43 BM samples and 65 PB specimens, an additional PE-conjugated anti-FcεRI antibody reagent (clone: AER-37) was also added. All antibody conjugated reagents were purchased from Becton/Dickinson Biosciences (BD; San Jose, CA), except for the CD45-PacO (Invitrogen; Carlsbad, CA, USA), CD36-FITC (Cytognos; Salamanca, Spain), anti-Slan-PE (Milteny Biotech; Cologne, Germany), CD300e and anti-FcεRI (Immunostep; Salamanca, Spain) antibodies. For sample preparation, the EuroFlow bulk lyse standard operating protocol (SOP) was used as previously described in detail [46,47]. Briefly, bulk erythrocyte lysis with ammonium chloride was performed. Subsequently, nucleated cells were washed and placed in phosphate-buffered saline solution (PBS) at a concentration of 10^7^ cells/200 µL. Concentrated 10^7^ BM and PB nucleated cells were then stained, washed and fixed with FACS lysing solution (BD) as per the EuroFlow SOP for staining of cell surface-only markers [46,47], available at www.EuroFlow.org (accessed on 4 May 2018). Stained samples were immediately measured in an LSR Fortessa X-20 flow cytometer (BD) using the FACSDiva^TM^ software (BD). Prior to acquisition, instrument setup, calibration and monitoring were performed using the EuroFlow instrument setup and compensation SOP for 12-color measurements [46] available at www.EuroFlow.org. For data analysis, the Infinicyt^TM^ software (Cytognos S.L) was employed. For every sample, the following subsets of Mo were systematically identified: (1) CD62L^+^ and, (2) CD62L^−^ cMo (CD14^hi^ CD16^−^); (3) iMo (CD14^hi^CD16^+^); and (4) CD36^+^Slan^−^, (5) CD36^−^Slan^−^, (6) CD36^+^Slan^+^, and (7) CD36^−^Slan^+^ ncMo (CD14^−/lo^CD16^+^). An additional population of FcεRI^+^ cMo (CD14^hi^CD16^−^) was also identified in the subset of samples stained with anti-FcεRI (*n* = 108). For the identification and enumeration of Mo and monocyte subsets, a previously defined gating strategy was used, as illustrated in Figure 1 [48].

### 2.3. Quantification of Soluble Cytokine plasma Levels Using the Cytometric Bead Array Platform

Soluble IL1β, IL6, IL8, IL10, IL12p70 and TNF-α levels were measured in a subset of 40 (IL1β, IL8 and IL12p70) to 50 (IL6 and TNF-α) freshly-frozen plasma samples from an identical number of subjects from the same patient cohort. The Cytometric Bead Array immunoassay (CBA) and the human inflammatory cytokine kit (BD) were used to quantify soluble IL1β, IL6, IL8, IL10, IL12p70 and TNF-α plasma levels, strictly following the recommendations of the manufacturer. Briefly, 50 μL of thawed plasma samples were incubated for 1 h at room temperature (RT) with 50 μL of a mixture of each of the anti-cytokine antibody-coated beads. Afterward, 50 μL of the PE-conjugated detection antibody reagent was added to each sample, followed by incubation for 2 h at RT. Once this incubation was completed, the unbound antibody was washed out (1×) and the washed beads were (immediately) measured in a FACSCanto II flow cytometer (BD). Data on 3000 events per bead population per sample were measured and stored for a total of 30,000 beads. For data analysis, the FCAP Array v3 (BD) and the CBA (BD) software programs were used, as described elsewhere [49]. The limit of quantitation (LOQ) for the distinct cytokines evaluated was as follows: IL1β, 7.2 pg/mL; IL6, 2.5 pg/mL; IL8, 3.6 pg/mL; IL10, 3.3 pg/mL; IL12p70, 1.9 pg/mL; TFN-α, 3.7 pg/mL.

### 2.4. ELISA (Enzyme-Linked Immunosorbent Assay) Quantitation of Bone-Derived Markers in Plasma

Quantitation of the soluble levels of BALP, RANKL and OPG was performed in a total of 40 plasma samples from an identical number of subjects from the same patient cohort using commercially available ELISA kits, according to the instructions of the manufacturer (Cusabio Biotech, Wuhan, China). After sample preparation, optical densities were read at 450 nm using a Tecan Spectra Fluor Plus™ microplate reader, and data were analyzed using the Curve Expert 1.4 software (Cusabio Biotech).

### 2.5. Statistical Methods

The Kolmogorov–Smirnov (KS) test was used to test the normality of data. Since data were not normally distributed, the non-parametric statistical Wilcoxon or the Friedman tests and the Mann–Whitney U or the Kruskal–Wallis tests were subsequently used to assess the (two-sided) statistical significance of differences observed between two or more than two groups for paired and unpaired variables, respectively. The Chi-square test was used to compare frequencies of cases between different groups. For multivariate analysis, the T-distributed stochastic neighbor embedding analysis (T-SNE) was used (Infinicyt^TM^ software). Statistical significance was set at *p*-values ≤ 0.05.

## 3. Results

### 3.1. Distribution of Monocytic Precursors, Mo, and Monocyte Subsets in BM

The distribution of monocytic precursors, as well as mature Mo and their subsets was first analyzed in BM aspirated samples of 19 MGUS, 13 SMM and 81 MM patients vs. 15 HD. Overall, a slightly increased percentage of monocytic precursors was observed in MGUS (vs. HD)-median (range) of 1.2% (0.83–2.3) vs. 0.9% (0.6–1.6) in HD (*p* = 0.2)-. In contrast, compared to MGUS cases, both SMM-median (range) of 0.9% (0.3–1.0; *p* = 0.01)- and MM-median (range) of 0.8% (0.4–1.7; *p* = 0.04)- showed lower counts of monocytic precursors in BM (Figure 2A) associated with significantly decreased percentages-median (range)- of monoblasts in BM of SMM vs. HD −0.08% (0.01–0.4) vs. 0.24% (0.01–0.6), respectively (*p* = 0.03)- (Figure 2B) and of promonocytes compared to MGUS −0.6% (0.1–3.7) vs. 1.1% (0.2–2.4), respectively (*p* = 0.03)- (Figure 2C). In line with these findings, the overall percentage-median (range)- of more mature Mo was significantly increased in BM of MGUS cases (vs. HD) −4.4% (2.2–7.8) vs. 3.5% (2.7–6.2) in HD (*p* = 0.01) but not of SMM −3.4% (1.3–6.2) vs. 4.4% (2.2–7.8) in MGUS (*p* = 0.02)- and MM cases −3.3% (1–6) vs. 4.4% (2.2–7.8) in MGUS (*p* = 0.003)- (Figure 2D). Such increased numbers of mature Mo in BM of MGUS was at the expense of the major compartment of recently produced cMo-median (range) of 3.6% (1.9–4.4) in MGUS vs. 2.4% (1.4–3.3) in HD (*p* = 0.005), 2.4% (1.2–3.4) in SMM (*p* = 0.02) and 2.2% (0.6–5) in MM (*p* = 0.005), respectively- (Figure 2E), altough no differences were observed among cMo subsets (Figure 2F–H). In contrast, MGUS, SMM and MM patients showed greater percentages -median (range)- in BM (vs HD) of both iMo −0.11% (0.03–0.41), 0.11% (0.04–0.2), and 0.11% (0.03–0.46) vs. 0.04% (0.0–1.1), respectively (*p* ≤ 0.02)- (Figure 2I) and of ncMo −0.17% (0.02–0.59), 0.17% (0.09–0.39), and 0.19% (0.03–0.57) vs. 0.07% (0.03–0.13), respectively (*p*≤0.03)- (Figure 2J). The later increase in ncMo -median (range)- observed in BM of MGUS, SMM and MM was at the expense of the CD36^+^Slan^−^ subset of ncMo -0.08% (0.01–0.13), 0.06% (0.02–0.15) and 0.09% (0.02–0.42) vs. 0.03% (0–0.04) in HD, respectively (*p*≤0.01)- (Figure 2K) alone in MGUS or in combination with that of CD36^−^Slan^−^ −0.05% (0.02–0.13) vs. 0.02% (0–0.05) in HD (*p* = 0.008)- and CD36^−^Slan^+^ ncMo -0.04% (0.01–0.09) vs. 0.01% (0–0.05) in HD (*p* = 0.03)- in SMM or that of the CD36^−^Slan^−^ -0.05 (0.01–0.24) vs. 0.02% (0–0.05) in HD (*p* = 0.005)- and CD36^+^Slan^+^ subsets of ncMo -0.01% (0.0–0.09) vs. 0.0% (0.0–0.03) in HD (*p* = 0.02)- in MM (Figure 2L–N).

Subsequently, the distribution of monocytic precursors and mature Mo and their subsets was investigated in active MM cases divided according to the international staging system (ISS) and the revised ISS (RISS). Thus, a decreased percentage -median (range)- of monoblasts was observed in BM of ISS-III and RISS-III patients -0.09% (0.01–0.5%) and 0.09% (0.01–0.3%)- vs. ISS-I and RISS-I cases -0.2% (0.03–0.6%) and 0.2% (0.01–0.6%), respectively (*p* ≤ 0.05)-. In turn, CD36^+^Slan^+^ ncMo were significantly increased in ISS-III vs. ISS-I patients -median (range) of 0.01% (<0.01–0.2%) vs. 0.01% (<0.01–0.02%; *p* = 0.05)- and FcεRI^+^ cMo were found at higher median (range) counts among RISS-III vs. RISS-II patients −0.4% (0.2–0.6%) vs. 0.05% (0.01–0.4%) *p* = 0.02-.

### 3.2. Distribution of Mo and Their Subsets in Blood

The overall distribution of Mo and their distinct subsets was subsequently investigated in blood of 22 MGUS, 13 SMM, and 88 MM patients, in parallel to 97 age-matched HD. Both SMM and MM patients displayed decreased -median (range)- counts in blood (vs HD) of total Mo −260 (98–950) cells/µL and 267 (71–1772) cells/µL vs. 388 (112–1131) cells/µL, respectively (*p* ≤ 0.05)- and of cMo −194 (71–756) cells/µL and 199 (32–1271) cells/µL vs. 297 (95–1052) cMo/µL, respectively (*p* ≤ 0.05)- (Figure 3A,B). In contrast to SMM and MM patients, MGUS cases displayed overall normal total Mo (and monocyte subsets) counts in blood (Figure 3). Of note, while decreased -median (range)- counts in blood of cMo in SMM were (exclusively) at the expense of CD62L^+^ cMo −141 (53–518) cells/µL vs. 230 (69–732) cells/µL in HD (*p* = 0.004)-, in symptomatic MM all subsets of cMo were significantly decreased vs. HD, including CD62L^+^ cMo −153 (59–476) cells/µL vs. 230 (69–732) cells/µL, respectively (*p* = 0.02)-, CD62L^−^ cMo −32 (4–131) cells/µL vs. 51 (9–189) cells/µL, respectively (*p* = 0.02)- and particularly, FcεRI^+^ cMo −1.4 (0.4–33) cells/µL vs. 30 (3–125) cells/µL, respectively (*p* = 0.001)- (Figure 3C–E).

In contrast to cMo, no significant differences were observed between the different patient groups and HD as regards both iMo and ncMo counts in blood (Figure 3F–K), except for decreased ncMo -median (range)- levels in MM -23 (4–71) cells/µL vs. 37 (11–84) cells/µL (*p* = 0.01)- (Figure 3G), at the expense of the CD36^−^Slan^−^, CD36^+^Slan^+^ and CD36^−^Slan^+^ subsets of ncMo −7 (1–31) cells/µL, 0.6 (0–5) cells/µL and 3 (0–18) cells/µL vs. 11 (2–43) cells/µL, 1.8 (0.3–5) cells/µL and 9 (3–24) cells/µL in HD, respectively (*p* ≤ 0.05)-. No statistically significant differences were observed in the distribution of CD36^+^Slan^−^ ncMo in blood among the different groups of patients (Figure 2H–K).

Among active MM patients, ISS-III and ISS-II cases showed higher median (range) counts of iMo and CD36^−^Slan^+^ ncMo −16 (0.8–232) and 13 (0–80) cells/µL- than ISS-I MM patients −8 (3–40) and 8 (0–46) cells/µL, respectively (*p* ≤ 0.02)-. In turn, RISS-III patients showed significantly higher counts of ncMo -median (range) of 50 (0.05–100) cells/µL- in blood, at the expense of CD36^−^Slan^−^ ncMo -median (range) of 17 (0.0–46) cells/µL-, compared to RISS-II cases -median (range) of 18 (0.0–195) and 9 (0.0–56) cells/µL, respectively (*p* ≤ 0.03)-.

### 3.3. Inflammatory Cytokine and Bone-Derived Marker Levels in Plasma

Overall, distinct immune/bone marker profiles were observed in the plasma of MGUS (*n* = 12), SMM (*n* = 7) and MM (*n* = 18) patients investigated (Figure 4). Thus, MGUS and SMM patients showed increased IL1β levels -median (range)- in plasma −2.3 (0.0–14) and 4.1 (0.0–9) pg/mL vs. undetectable levels in HD, (*p* = 0.007 and *p* = 0.01), respectively-, while these were within the normal range (e.g., usually undetected) in MM −0.0 (0.0–21) pg/mL, vs. HD *p* = 0.3-. In contrast, greater IL8 serum levels were observed in both SMM and MM, -median (range) of 115 (13–2013) and 54 (0.0–1071) vs. 14 (5–29) pg/mL in HD, respectively (*p* ≤ 0.01), but not in MGUS (*p* = 0.13). Interestingly, while MM patients showed increased -median (range)- IL6 −21 (0–133) pg/mL-, BALP −484 (200–1352) ng/mL- and RANKL −985 (498–4246) pg/mL- levels in plasma, these three later markers were within the normal range in plasma of both SMM −1.2 (0.08–9) pg/mL, 159 (29–366) ng/mL and 453 (109–858) pg/mL vs. 2.5 (0–22) pg/mL, 173 (120–316) ng/mL and 580 (547–1082) pg/mL in HD, respectively (*p* > 0.05)- and MGUS (Figure 4A–G). Interestingly, when we compared MM patients with and without osteolytic lesions at diagnosis, significantly higher levels of IL6 -median (range) of 48 (0–32) vs. 13 (13–400) pg/mL, respectively (*p* = 0.001)- and BALP -median (range) of 594 (211–4201) vs. 387 (197–739) pg/mL, respectively (*p* = 0.05)- were detected among MM patients that had osteolytic lesions. Either normal (i.e., OPG) or undetectable (i.e., IL10, IL12p70 and TNFα) levels were found in the plasma of MGUS, SMM and MM patients for the other cytokines and bone-associated markers investigated (Figure 4F). Despite this, an increased RANKL/OPG ratio was observed in MM (either in presence or absence of bone osteolytic lesions) vs. SMM patients -median (range) of 3 (0.8–4) vs. 1 (0.2–2), *p* ≤ 0.01- (Figure 4G). Of note, no significant correlation was observed between the Mo subset counts and inflammatory cytokine or bone-derived marker levels in plasma which might be due to the relatively small number of samples in which both setoff markers were analyzed.

Based on these findings, multivariate T-SNE analysis showed three clearly different patient clusters/groups (*p* = 0.0001) (Figure 4H). A first cluster/group consisted of all HD (50%) and most MGUS (30%) patients together with a smaller fraction of (all low-risk) SMM cases (20%). In a second group, SMM (including one-third of high-risk SMM cases) predominated (60%), together with a small fraction of MGUS (20%) and MM (20%) patients who showed no osteolytic lesions. Finally, the third group exclusively consisted of MM patients (100%), most of whom (97%) had osteolytic lesions at diagnosis.

Of note, all patients in group 1 showed normal serum levels of the different cytokine and bone markers investigated (Figure 5). In contrast, group 3 (MM) cases systematically showed increased -median (range)- levels of IL6 −41 (13–444) pg/mL-, BALP −656 (367–4201) ng/mL- and RANKL −1497 (841–4246) pg/mL- when compared with both group 1 −2.3 (0–22) pg/mL (*p* = 0.0001), 170 (29–316) ng/mL (*p* = 0.0001) and 533 (109–1082) pg/mL (*p* = 0.0001), respectively- and group 2 cases −3 (0.1–12) pg/mL (*p* = 0.0007), 245 (159–408) ng/mL (*p* = 0.005) and 737 (434–858) pg/mL (*p* = 0.001), respectively- (Figure 5). In turn, patients in group 2 showed greater levels -median (range)- in plasma of IL1β −6 (2–9) vs. 0.0 (0.0–4) and 0.0 (0.0–11) pg/mL in group 1 (*p* = 0.0003) and group 3 (*p* = 0.002), respectively- and IL8 −1071 (152–2013) vs. 21 (4–72) and 62 (44–203) pg/mL in group 1 (*p* = 0.0002) and in group 3 (*p* = 0.005), respectively- (Figure 5). Of note, MM patients included in group 3 also showed significantly lower counts -median (range)- of classical FcεRI^+^ Mo in blood −6.7 (0.5–16) vs. 49 (15–97) cells/µL in group 1 (*p* = 0.01) and 30 (3–62) cells/µL in group 2 cases (*p* = 0.2)- (Figure 5G) while a similar distribution was observed in blood and BM among the three groups for all other subsets of mature Mo investigated (Appendix A). Despite this, group 2 patients showed lower percentages of monocytic precursors −0.36% (0.1–0.9) vs. 1.0% (0.6–4) and 0.8% (0.4–1.2) in group 1 (*p* = 0.02) and group 3 (*p* = 0.1), respectively-, including both lower counts of monoblasts −0.04% (0.01–0.2) vs. 0.1% (0.07–0.4) and 0.09% (0.06–0.3) in group 1 (*p* = 0.05) and 3 (*p* = 0.04), respectively- and promonocytes −0.32% (0.1–0.7) vs. 0.8% (0.5–4) and 0.7% (0.3–0.9) in group 1 (*p* = 0.01) and 3 (*p* = 0.2), respectively- (Appendix A).

## 4. Discussion

At present, it is well-established that both the immune and bone microenvironment play a key role in the pathogenesis of PC neoplasms [4,5]. Among other immune cells [30,31,32,33,34,35,36,37,38,39,40,41,42,43,44,45], Mo [25,26,50] and macrophages [6,7,9] have emerged as key players in disrupting normal BM and bone homeostasis. This translates into increased levels in serum of multiple inflammatory cytokines [27,28] and markers [11,12,13,14,15,16,17,18,21,22] associated with increased bone resorption, particularly in MM [4,5]. In parallel, important advances have been achieved in the understanding of the maturation pathways leading to the production of Mo in BM and their heterogeneity in blood and BM, including the functional role of the many subsets of Mo identified so far [48,51,52,53,54,55,56,57,58]. Despite this, at present, there are limited data about the distribution of the distinct subsets of Mo in blood and BM of MGUS, SMM and MM patients, and the potential relationship with serum inflammatory and bone-associated biomarkers, known to be altered in these patients [11,12,13,14,15,16,17,18,21,22,27,28]. Here, we first investigated the distribution of Mo and monocyte subsets in blood and BM of a large cohort of MGUS, SMM and MM patients compared with age-matched HD. Subsequently, we determined the potential associations between the alterations identified in the blood and BM monocyte compartment and serum inflammatory and bone-associated biomarker profiles in a subset of the patients.

Overall, our results showed a progressively decreased production of monocytic precursors in both SMM and MM, but not MGUS. This translated into decreased counts of cMo in BM associated with lower counts of either recently produced CD62L^+^ cMo in the blood of SMM and of all subsets of (CD62L^+^, CD62L^−^ and FcεRI^+^) cMo in blood of MM, while MGUS cases showed a normal (or even slightly increased) production of cMo. In contrast, the more mature/differentiated iMo and end-stage ncMo were significantly increased in BM of both SMM and MM as well as MGUS patients, while their counts in blood were within the normal range (MGUS and SMM) or decreased (MM). Of note, among MM patients, cases at more advanced stages of the disease (e.g., ISS-III and RISS-III) showed lower percentages of monoblasts than ISS-I and RISS-I MM patients, together with higher percentages of CD36^+^Slan^+^ ncMo in BM. Interestingly, FcεRI^+^ cMo were significantly more represented in BM of RISS-III vs. RISS-II. In addition, RISS-III cases also showed higher absolute counts of ncMo (at the expense of CD36^−^Slan^−^ ncMo). In blood, ISS-III or ISS-II MM cases had higher counts of blood circulating iMo and CD36^−^Slan^+^ ncMo vs. ISS-I MM cases.

Altogether, the above alterations might reflect early activation and increased production of Mo in MGUS, followed by chronic inflammation with a local increase in BM of more mature iMo and functionally exhausted ncMo. Subsequently, disruption of monocytic production in BM of SMM and MM patients would lead to a deficient production and release of recently formed CD62L^+^ cMo into blood and thereby, decreased numbers of circulating cMo. Growth of the tumor PC compartment in BM would progressively favor a sustained increase of iMo and end-stage (functionally exhausted) inflammatory Slan^+^ ncMo, together with inhibition in symptomatic MM, of the generation and/or accumulation of other subsets of CD62L^−^, and particularly FcεRI^+^ cMo. These results confirm and extend on previous observations that revealed quantitative alterations of cMo, iMo and ncMo, as well as of end-stage (Slan^+^) ncMo in blood and BM of MGUS and MM [19,50]. In line with these findings, previous studies have shown increased numbers of iMo in BM of MM and demonstrated their pro-osteoclastogenic potential in ex vivo cultures. Altogether these findings highlight the potential relevance of iMo in the pathogenesis of bone disease in MM [19,59]. Despite this, these findings should be taken with caution and deserve further confirmation, as BM analyses were conducted in BM aspirated samples and not directly in core (biopsy) BM tissue specimens.

In order to better understand the functional implications of these (altered) patterns of monocyte production and maturation kinetics in the different BM and blood compartments, we further investigated the levels of several inflammatory cytokines in the serum of our patients. Overall, our results showed increased levels of IL1β, IL6, and/or IL8 in a fraction of MGUS and all SMM and MM patients. In line with these findings, Martín-Ayuso et al. [28] have previously shown enhanced ex vivo secretion (vs HD) of proinflammatory cytokines, such as IL6, IL8, IL12, TNFα and IL15 by DC and ncMo in MGUS and MM patients in the presence of undetectable IL1β serum levels. Similarly, Bosseboeuf et al. [60] found increased levels (vs HD) of serum IL1β, IL6 and IL8 (out of 40 cytokines investigated) in a pooled series of 64 MGUS plus MM patients. More interestingly, when we considered MGUS, SMM and MM patients separately, increased serum IL1β was specifically found among MGUS and SMM patients, while increased serum IL8 levels were typically detected in SMM and MM cases, and elevated serum IL6 was almost restricted to MM, supporting the existence of different patterns of secretion of inflammatory cytokines in MGUS, SMM and MM. Of note, IL1β and IL8 have both been linked to tumor cell- and immune cell-associated senescence, and acquisition of a pro-inflammatory secretory phenotype (SASP), which would favor further genetic and chromosomal instability, with relatively limited effects on bone metabolism [61]. In contrast, previous studies have shown that IL6 is a proinflammatory cytokine also involved in bone remodeling [20], that once released to the bone matrix inhibits osteoblastic activity and induces production of RANKL, parathormone-related protein and prostaglandin E2, with a (synergistic) pro-osteoclastic effect with IL6 in promoting bone resorption [20]. Local production of parathormone and vitamin D further stimulate osteoblasts to produce more IL6 and RANKL in a positive feedback loop that further enhances osteoclastogenesis [20]. In line with these findings, our results showed a parallel increase in serum IL6, RANKL and BALP [17,18] restricted to MM, particularly to those MM patients who showed osteolytic lesions at diagnosis. However, in contrast with several previous studies, we could not confirm a simultaneous decrease in OPG serum levels in MM [14,15]. Despite this, the RANKL/OPG ratio was significantly increased in MM vs. both MGUS and SMM. Of note, despite the macrophage inflammatory protein (MIP)-1α also plays an important role in osteoclast formation with a potential for further enhancing RANKL and IL6 effects on osteoclasts, in this study we did not analyze MIP 1α levels in plasma [62]. In any case, based on the plasma levels of the IL1β, IL8 and IL6 cytokines and the BALP and RANKL bone-related markers, three clearly different profiles were identified among HD, and MGUS, SMM and MM patients. Thus, HD, most MGUS cases and a minority of SMM (low-risk)showed a normal cytokine and bone-marker serum profile, while a senescent-like pattern was observed among the remaining MGUS cases, most SMM patients and a minority of MM who showed no bone lesions, and a pro-inflammatory profile was restricted to MM patients who typically had osteolytic lesions. Of note, the later profile was associated with significantly decreased numbers of the FcεRI^+^ subset of cMo in blood. In this regard, recent studies have shown that FcεRI^+^ cMo are a functionally different subset of Mo with a unique immunomodulatory role, which act as the regulator of (allergic and potentially also other types of) inflammation [63,64]. If this holds true, an impaired function of this specific subset of immunomodulatory Mo might facilitate the transition in MM between a senescent-like and a more prominent pro-inflammatory (IL6-associated) microenvironment, associated with progressive emergence of bone disease [65]. However, further studies are required to establish the actual functional role of these cells in humans, and their potentially immunomodulatory effect in MGUS and SMM vs. MM.

## 5. Conclusions

Our results show an altered distribution of Mo subsets in BM of MGUS, SMM and MM, associated with a progressively decreased production of CD62L^+^ cMo in SMM (blood), in addition to CD62L^−^ (blood and BM) and FcεRI^+^ cMo in MM (blood).

## Figures and Tables

**Figure 1 cancers-13-01454-f001:**
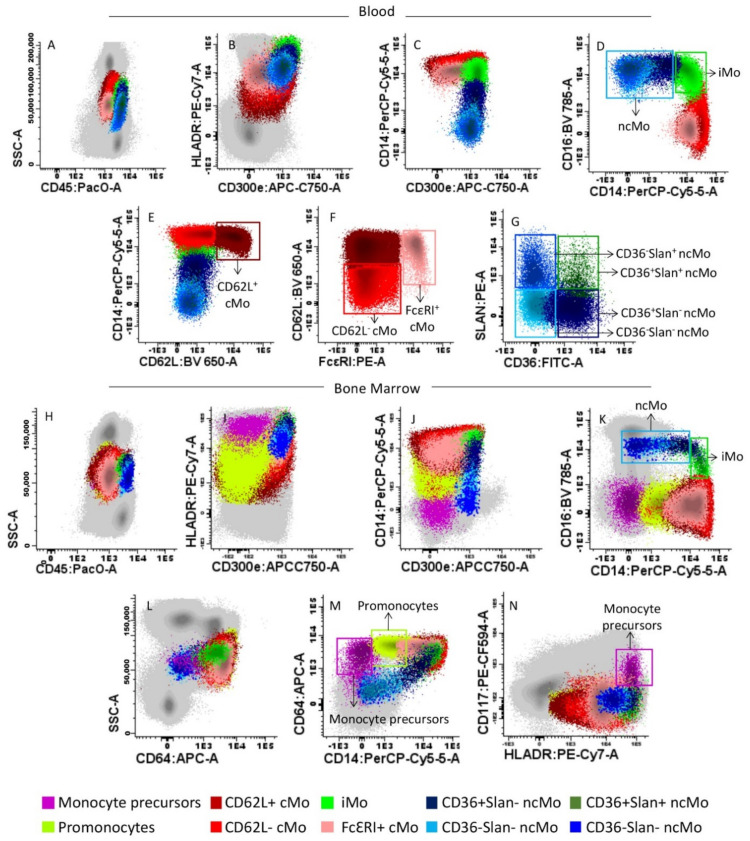
Gating strategy used for the identification of monocytes (Mo) and their different subsets in representative peripheral blood (PB) (upper panels **A**–**G**) and bone marrow (BM) (lower panels **H**–**N**) samples from a healthy donor (HD). The distinct blood (panels **A**–**G**) and BM (panels **H**–**N**) subsets of Mo identified are represented as colored events, after they had been gated and discriminated from all other PB and BM leucocytes (gray dots) based on their typical SSC/CD45 profile (panels **A**,**H**), and their unique positivity for CD300e, HLADR, CD64 and CD14 (panels **B**,**C**,**I**,**J**,**M**). Within total (mature) Mo, the following subsets were identified based on their distinct expression profile for CD14 and CD16 (panels **D**,**K**) plus FcεRI (panel **F**): CD14^hi^CD16^−^ classical (c)Mo, CD14^hi^CD16^+^ intermediate (i) Mo, and CD14^−/lo^CD16^+^ non-classical (nc)Mo and the subset of CD14^+^ CD16^−^ FcεRI^+^ cMo. Within cMo, the CD62L^+^ and CD62L^−^ subsets were further identified (panel **E**), while ncMo were further separated into CD36^+^Slan^−^, CD36^−^Slan^−^, CD36^+^Slan^+^ and CD36^−^Slan+ ncMo (Panel **G**). In BM samples, monoblasts and promonocytes were additionally identified, based on their phenotypic profile (positivity) for both CD117, HLADR (panels **N**) and expression of CD64 (panel **L**,**M**).

**Figure 2 cancers-13-01454-f002:**
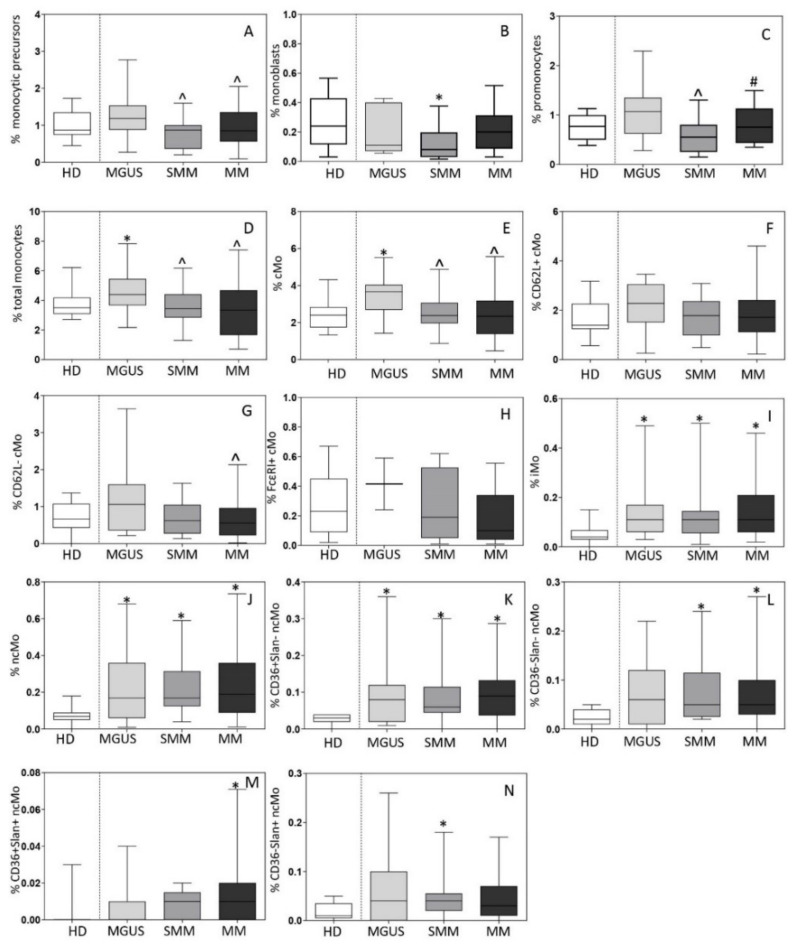
Distribution of monocytic precursors (total monocytic precursors, monoblasts and promonocytes; panels **A**–**C**), total monocytes (Mo) (panel **D**) and their subsets (panels **E**–**N**) in bone marrow (BM) of newly-diagnosed MGUS (*n* = 19), SMM (*n* = 13) and MM (*n* = 81) patients vs. age-matched HD (*n* = 15). In panels A-N, notched boxes extend from the 25th to the 75th percentile values; the lines in the middle and vertical lines correspond to median values and the 5th and 95th percentiles, respectively. Statistical significant differences (*p* ≤ 0.05) were observed vs. * HD, ^ MGUS, and # SMM patients after applying the Mann–Whitney non-parametric test (as the KS normality test showed that data for these variables did not follow a normal distribution). cMo: classical monocytes; iMo: intermediate monocytes; ncMo: non-classical monocytes; HD: healthy donor; MGUS: monoclonal gammopathy of undetermined significance; SMM: smoldering multiple myeloma; MM: multiple myeloma.

**Figure 3 cancers-13-01454-f003:**
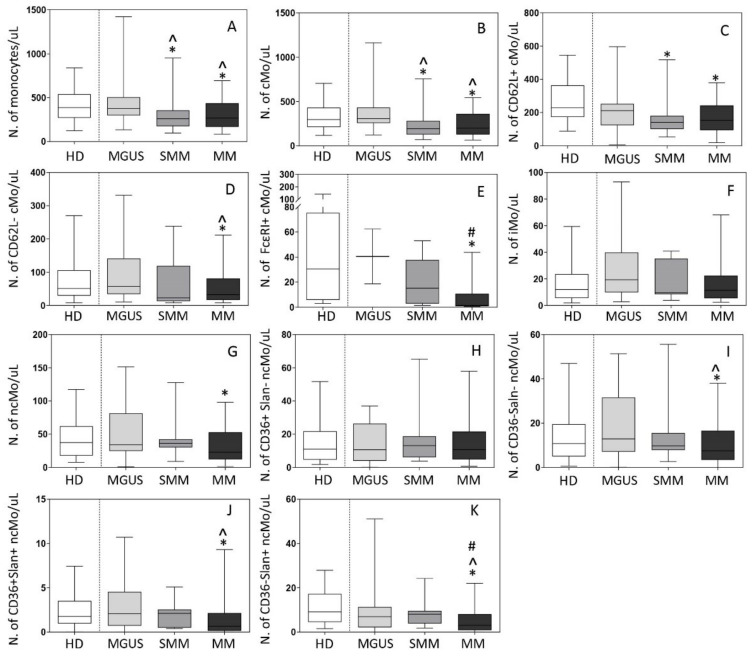
Distribution of total monocytes (Mo) (panel **A**) and monocyte subsets (panels **B**–**K**) in blood of newly-diagnosed MGUS (*n* = 22), SMM (*n* = 13) and MM (*n* = 88) patients vs. age-matched HD (*n* = 97). In (panels **A**–**K**), notched boxes extend from the 25th to the 75th percentile values; the lines in the middle and vertical lines correspond to median values and the 5th and 95th percentiles, respectively. Statistical significant differences (*p* ≤ 0.05) were observed vs. * HD, ^ MGUS, and # SMM patients after applying the Mann–Whitney non-parametric test (as the KS normality test showed that data for these variables did not follow a normal distribution). cMo: classical monocytes; iMo: intermediate monocytes; ncMo: non-classical monocytes; HD: healthy donor; MGUS: monoclonal gammopathy of undetermined significance; SMM: smoldering multiple myeloma; MM: multiple myeloma.

**Figure 4 cancers-13-01454-f004:**
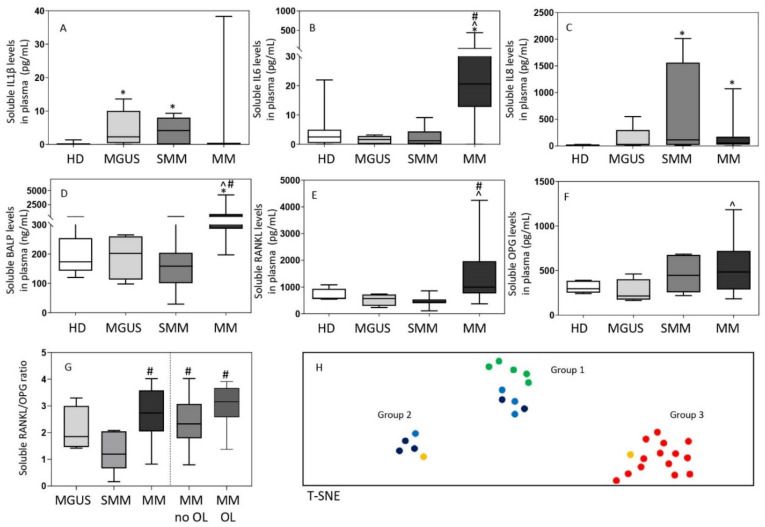
Soluble levels of cytokines (panels **A**–**C**) and bone-associated markers (panels **D**–**G**) and the overall immune/bone profiles (panel **H**) identified in plasma of MGUS (*n* = 12), SMM (*n* = 7), MM (*n* = 18) and HD (*n* = 13). In panels A-G, notched boxes extend from the 25th to the 75th percentile values; the lines in the middle and vertical lines correspond to median values and the 50th and 95th percentiles, respectively. In panel H, HD are depicted as green dots, while MGUS, SMM and MM are colored as light blue, dark blue, and either yellow (no osteolytic lesions at diagnosis) or red (osteolytic lesions) dots, respectively. Statistical significant differences (*p* ≤ 0.05) were observed vs. * HD, ^ MGUS, and # SMM patients after applying the Mann–Whitney non-parametric test (as the KS normality test showed that data for these variables did not follow a normal distribution). IL: interleukin; BALP: bone alkaline phosphatase; RANKL: receptor activator of nuclear factor-kB ligand; OPG: osteoprotegerin; HD: healthy donors; MGUS: monoclonal gammopathy of undetermined significance; SMM: smoldering myeloma; MM: multiple myeloma; OL: osteolytic lesions; T-SNE: T-distributed stochastic neighbor embedding.

**Figure 5 cancers-13-01454-f005:**
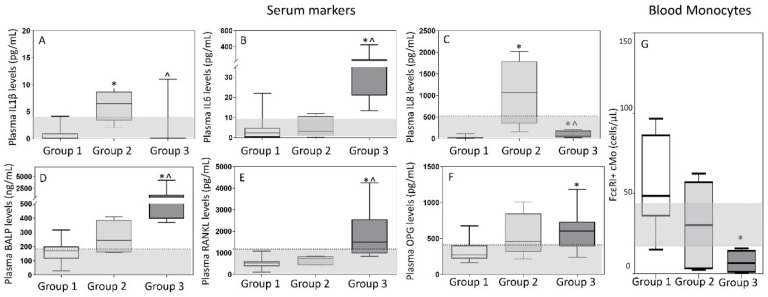
Soluble levels of cytokines (panels **A**–**C**) and bone-associated markers (panels **D**–**F**) in plasma and absolute FcεRI^+^ cMo counts in blood (panel **G**) of MGUS, SMM, MM and HD classified by multivariate analysis into the T-distributed stochastic neighbor embedding (T-SNE)-defined groups 1, 2 and 3. In (panels **A**–**G**), notched boxes extend from the 25th to the 75th percentile values; the lines in the middle and vertical lines correspond to median values and the 5th and 95th percentiles, respectively. Normal range is represented by a gray area limited by a horizontal black dotted line. Statistical significant differences (*p* ≤ 0.05) were observed vs. * group 1 or ^ group 2 after applying the Mann–Whitney non-parametric test (as the KS normality test showed that data for these variables did not follow a normal distribution). IL: interleukin; BALP: bone alkaline phosphatase; RANKL: receptor activator of the nuclear factor-kB ligand; OPG: osteoprotegerin; cMo: classical monocytes.

## Data Availability

The data presented in this study are available on request from the corresponding author.

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
