# Peer review of "Monocyte Subsets and Serum Inflammatory and Bone-Associated Markers in Monoclonal Gammopathy of Undetermined Significance and Multiple Myeloma"

_cancers, 2021, doi:10.3390/cancers13061454_

Round 1

Reviewer 1 Report

The authors here investigate the distribution of different subsets of monocytes in blood and bone marrow of monoclonal gammopathy of undetermined significance (MGUS), smoldering (SMM) and symptomatic multiple myeloma (MM) and its relationship with immune/bone serum-markers profiles.

There are few data about the distribution of the distinct subsets of monocytes in blood of myeloma so it is an original study.

The paper is well written but I do have comments and suggestions for improvement.

  1. Material and methods. In this section the population of healthy donors (n=107) was not well characterized. How healthy donors have been recruited ? Healthy blood donors ? Healthy donors from general population ?

  1. Material and methods. What imaging did the authors use to diagnose osteolytic lesions in multiple myeloma patients ? Conventional skeletal survey or whole-body CT which has been shown to have superior sensitivity in detecting osteolytic lesions and is considered standard of care to diagnose the myeloma bone disease.

  1. Quantification of cytokine plasma levels. Among this large panel of cytokines, why the authors did not measure MIP-1a which is a potent cytokine that induces osteoclast formation while it potentiates RANKL and Il-6 effects on osteoclasts ?

  1. ELISA quantitation of bone-derived markers in plasma. Would consider analysis of RANKL/OPG ratio which is an independent pronostic factor in newly diagnosed myeloma patients.

  1. Supplementary Table 1. I will consider adding Revised International Staging System (R-ISS)

  1. I would suggest to investigate and compare the distribution of monocytes subsets in subgroups of symptomatic multiple myeloma patients according to ISS and R-ISS.

  1. Monocytes are involved in pathogenesis of myeloma bone disease with the release of mediators that promote osteoclastogenesis. Non classical monocytes may be markers of circulating osteoclast precursors. I will suggest adding 2 references below :

Petitprez V, Royer B, et al. CD14+ CD16+ monocytes rather than CD14+ CD51/61+ monocytes are a potential cytological marker of circulating osteoclast precursors in multiple myeloma. A preliminary study. Int J Lab Hematol. 2015;37(1):29–35. doi: 10.1111/ijlh.12216.

Bolzoni M, Ronchetti D, et al. IL21R expressing CD14(+)CD16(+) monocytes expand in multiple myeloma patients leading to increased osteoclasts. Haematologica. 2017;102(4):773–784. doi: 10.3324/haematol.2016.153841. 

Reviewer 2 Report

Well done.

Reviewer 3 Report

This is a very interesting manuscript in which authors characterize  different monocyte populations in blood  and bone marrow  samples of a large cohort of patients at the three stages of multiple myeloma progression (MGUS, Smoldering and MM) vs healthy controls. Authors find an altered distribution of Mo subsets in MGUS,  SMM and MM, associated with a progressively decreased production of CD62L+ cMo in SMM (blood) in addition to CD62L- (bm and blood) and FcεRI+ cMo (blood) in MM.

They also correlate data with immune/bone serum markers. Authors identify three interesting distinct immune/bone  profiles: 1) normal (typical of controls and most MGUS cases); 2) senescent-like (found in a minority of MGUS, most SMM and few MM cases with no bone lesions); and 3) pro inflammatory (typical of MM presenting with bone lesions). Using these clusters, the authors found a correlation with decreased counts in blood of immunomodulatory  FcεRI+cMo.

Specific comments:

-Table S1 do not specify whether some MM patients were treated before diagnosis (eg. from high-risk smoldering). Please comment

-It would be helpful, in order to better understand the analysis performed,  if table S1 also indicates the serum samples used in the study

-Are serum samples from the same individuals? If this, please indicate. Also, provide the number of subjects in heading 3.3, figure 4 and figure S1 and S2

-Authors do not describe results regarding iMo and mcMo populations in BM  which are increased when compared to controls (but not as MM progress). Please comment

-It would be very interesting if MM data were subdivided in MM with/without osteolytic lesions. For example, it is IL-6 more represented in MM patients with osteolytic lesions?

-In figure 4D, 20% of the patients of group 1 are smoldering whereas in group 2 smoldering represent the 60% of patients. Have you correlated these data with low or high risk smoldering?

-Have you performed a correlation/ regression analysis between monocyte populations and serum markers?

Minor points:

-In conclusions, this statement describe which happens in bone marrow but also blood samples: our results show an altered distribution of Mo subsets in BM of MGUS,  SMM and MM, associated with a progressively decreased production of CD62L+ cMo in  SMM (blood), in addition to CD62L- (blood and bm) and FcεRI+ cMo in MM (blood)

-Please correct reference 50

Reviewer 4 Report

Summary : This study provides a thorough description of monocyte/macrophage markers in the plasma and bone marrow aspirates of    healthy donors and those with plasma cell dyscrasia.  The experimental design is fundamentally limiteds to observation and therefore cause and effect are entirely speculative. It is therefore of interest as a hypothesis-generating manuscript. 

Methodology : Section 2.1 : Please clarify what proportion of patients were paired between peripheral blood and marrow. Were any patients sequential in  moving from MGUS to SMM to MM?

Please emphasise that that is is bone marrow aspirate, which is subject to contamination by peripheral blood - a limitation of this sample type.

Section 2.5 - I am surprised the data are non-parametric or is this just a conservative approach given the small N? The specific statistical test used to obtain the p values Figures 2-5 should be made clear

Editing : Define HD Health Donors ins the introduction. It currently relies on seeing it in the abstract

Line 172 - 'a was used' - please clarify

Figure Box plots - I do not understand the rationale for the 'whiskers' of the plot. A 95% conifiendence interval would make more sense, especially with plotting of individual data points as the N are small. This would help for example in Figure 2I where the significance are hard to comprehend given the large overlapping error bars.

Round 2

Reviewer 3 Report

The authors have asked for all my suggestions and comments.